# Energy-Reduced Fabrication of Light-Frame Ceramic Honeycombs by Replication of Additive Manufactured Templates

**DOI:** 10.3390/ma16144924

**Published:** 2023-07-10

**Authors:** David Köllner, Sebastian Niedermeyer, Miklos Vermes, Swantje Simon, Ken-ichi Kakimoto, Tobias Fey

**Affiliations:** 1Department of Materials Science and Engineering (Institute of Glass and Ceramics), Friedrich-Alexander-Universität Erlangen-Nürnberg, Martensstr. 5, 91058 Erlangen, Germany; 2Frontier Research Institute for Materials Science, Nagoya Institute of Technology, Gokiso-cho, Showa-ku, Nagoya 466-8555, Japan

**Keywords:** ceramic honeycombs, replica method, additive manufacturing, piezoelectric porous ceramics, sol–gel method

## Abstract

Ceramic components require very high energy consumption due to synthesis, shaping, and thermal treatment. However, this study suggests that combining the sol–gel process, replica technology, and stereolithography has the potential to produce highly complex geometries with energy savings in each process step. We fabricated light-frame honeycombs of Al_2_O_3_, Ba_0.85_Ca_0.15_Zr_0.1_Ti_0.9_O_3_ (BCZT), and BaTiO_3_ (BT) using 3D-printed templates with varying structural angles between −30° and 30° and investigated their mechanical and piezoelectric properties. The Al_2_O_3_ honeycombs showed a maximum strength of approximately 6 MPa, while the BCZT and BaTiO_3_ honeycombs achieved a d_33_ above 180 pC/N. Additionally, the BCZT powder was prepared via a sol–gel process, and the impact of the calcination temperature on phase purity was analyzed. The results suggest that there is a large energy-saving potential for the synthesis of BCZT powder. Overall, this study provides valuable insights into the fabrication of complex ceramic structures with improved energy efficiency and enhancement of performance.

## 1. Introduction

Piezoelectric ceramic foams are highly attractive for sensing, actuation, and energy harvesting applications due to high sensitivity, high hydrostatic figure of merit (HFOM), low acoustic impedance, and improved signal-to-noise ratio [1,2,3,4,5]. Simultaneously, the mechanical properties decrease exponentially with increasing porosity, which limits them for particular applications [6,7,8]. In contrast to foams, metamaterials such as honeycombs show better mechanical properties with the same porosity. While Young’s modulus, Poisson’s ratio, and compressive strength can be adjusted via structural parameters [9,10,11], there is no general rule for their piezoelectric properties. Nevertheless, it was also shown that ceramic honeycombs can exhibit strain amplification and further increased piezoelectric constants [12].

Concurrently, ceramic honeycombs have highly anisotropic properties that vary greatly in-plane and out-of-plane [13,14] and are permeable in only one direction. However, an open-pore cell structure with permeability in all spatial directions would be preferred for applications such as hydrophones and medical implants [15,16,17]. For this reason, a new approach was investigated in this work by fabricating light-frame honeycombs with an open-walled cell structure by replication of additive manufactured (AM) templates, to benefit from the structural advantages of mechanical and piezoelectric properties while generating interconnectivity.

The piezoelectric properties will be also in focus since it is not clear how piezoelectric constants are related to the cellular structure of porous ceramics. For instance, ceramic foams with a random cell structure show a linear decrease in the piezoelectric constant d_33_ with increasing porosity [5,18,19,20], while freeze-cast ceramics with a lamellar or linear pore structure show irregular behaviors [21,22,23,24]. With further periodization and homogenization of the cell structure, it has even already been shown that, despite porosities of up to 60%, only a slight reduction in d_33_ can occur [25]. We are particularly interested in honeycombs because even increased values could be obtained compared to dense material [12,26]. For this reason, the light-frame honeycombs fabricated here can help to better understand how cellular structures can affect piezoelectric properties, as they are a mixture of different cellular types.

The traditional production of piezoelectric components is very energy-intensive since high synthesis, calcination, and sintering temperatures are necessary. Alternative synthesis routes, such as the sol–gel process, can reduce energy consumption. In the specific case of (BaCa)(ZrTi)O_3_ (BCZT), solid-state synthesis needs temperatures over 1300 °C [27,28], whereas sol–gel processes require less than 900 °C [29,30]. From flexible precursors, such as carbonates, acetates, alkoxides, hydroxides, and salts [29,30,31,32], the sol–gel process obtains high-purity nanopowders that can additionally reduce the sintering temperature. BCZT is additionally a promising candidate to replace lead-containing piezoceramics such as PZT in many commercial applications in the future, as it has already achieved better piezoelectric properties [27,33,34,35].

In the present work, light-frame Al_2_O_3_, BCZT, and BaTiO_3_ (BT) honeycombs were manufactured by replicating 3D-printed polymer templates with varying structural angles θ between −30° and 30°. The influence of the structure on the mechanical properties was investigated using a compression test and the piezoelectric properties via direct excitation. The used BCZT powder was prepared from a sol–gel route, and the effect of calcination temperature on phase purity was characterized with XRD and helium pycnometer.

## 2. Materials and Methods

### 2.1. Sol–Gel Synthesis

Ba_0.85_Ca_0.15_Zr_0.1_Ti_0.9_O_3_ (BCZT) powder was synthesized via a similar sol–gel route, according to Castkova et al. [29]. For solution A, barium carbonate (99%, Alfa Aesar, Haverhill, MA, USA) and calcium carbonate (≥99.0%, Sigma Aldrich, St. Louis, MI, USA) were dissolved in 5 M acetic acid under continuous stirring. For solution B, titanium(IV)-propoxide (97%, Sigma Aldrich), zirconium propoxide (70% in isopropyl alcohol, Alfa Aesar), and glacial acetic acid were added to ethanol under continuous stirring. Both solutions were mixed and aged overnight for a minimum of 12 h. The clear solution was heated until gelation and dried at 250 °C for 24 h. This xerogel was then calcined at 600 °C, 650 °C, 700 °C, 750 °C, and 800 °C for 5 h, respectively, to determine the appropriate calcination temperature.

### 2.2. Sample Preparation

Light-frame ceramic honeycombs with different angles θ were manufactured combining stereolithography (SLA) and replica techniques, described in detail elsewhere [25]. The auxetic (θ < 0°) and hexagonal (θ ≥ 0°) honeycomb unit cell templates are shown in Figure 1 and were created with the open-source CAD software OpenSCAD (version 2021.01) [36]. According to [25], the polymer struts exhibit a triangular shape to ensure a homogeneous coating. The templates were 3D printed with a Photon Mono X (Anycubic, Shenzhen, China) with the resin Formfutura Castable Wax (Formfutura Castable LCD series, Nijmegen, The Netherlands) with a layer resolution of 25 µm, cleaned in isopropanol, and cured for 6 min (Anycubic Wash & Cure Plus Machine, Anycubic, Shenzen, China). The polymer templates were impregnated three times with a BCZT, BT, or Al_2_O_3_ slurry, each time drying for 20 min at 80 °C in between. The composition of the BCZT and BT slurry was 21.3 vol.% BCZT or BT (Sigma-Aldrich Chemie GmbH, Taufkirchen, Germany), 74.8 vol.% distilled water as a solvent, 3.2 vol.% Darvan 821A (R.T. Vanderbilt Company, Inc., Norwalk, CT, USA) as a dispersant, 0.5 vol.% PEG 10,000 (Merck KGaA, Darmstadt, Germany) as a binder, and 0.2 vol.% Agitan 299 (Münzing Chemie GmbH, Heilbronn, Germany) as a defoaming agent. The Al_2_O_3_ slurry consists of 8.9 vol.% CT 3000 SG and 23.7 vol.% CT 3000 SDP (Almatis GmbH, Ludwigshafen, Germany), 62.3 vol.% distilled water, 0.3 vol.% citric acid (Carl Roth GmbH + Co.KG, Karlsruhe, Germany), 2.8 vol.% 2 M NaOH, 0.2 vol.% Agitan 299, and 2.0 vol.% PVA 8–88 (Sigma-Aldrich Chemie GmbH, Taufkirchen, Germany). As references, 30, 45, and 60 ppi polyurethane foams (Koepp Schaum, Germany) were used as templates to produce reference BCZT ceramic foams in the same porosity range, with the same method. The structure is shown in Figure 4d.

After drying for 24 h at RT, burnout of the organics was performed on porous annamullite substrates at up to 600 °C with heating rates (HR) between 0.1–5 K/min, followed by sintering at 1400 °C for 6 h (BCZT, BT) or 1600 °C for 2 h (Al_2_O_3_) (HR 5 K/min). All specimens were embedded in the two-component epoxy resin Technovit^®^ 5017 (Kulzer GmbH, Hanau, Germany) machined plane parallel with a pot grinder MPS 2 R220 (15 µm grain size, G&N GmbH, Erlangen, Germany), and finally released in acetone. The complete process scheme is shown in Figure 2.

### 2.3. Characterization

The crystal structures of the BCZT powders were examined by X-ray diffraction (D8 ADVANCE eco, Bruker Corporation, Billerica, MA, USA), and the pure densities with a helium pycnometer (AccuPyc II 1340, Micromeritics GmbH, Aachen, Germany). The specific surface area and particle size were determined by nitrogen absorption via the BET method (Gemini VII, Micromeritics GmbH, Aachen, Germany). The microstructures of the powders and cellular structures were investigated with scanning electron microscopy (Quanta 200 FEG, FEI Company, Hillsboro, OR, USA). DSC and TGA measurements were performed with a NETZSCH STA 449F3 (Selb, Germany) from 15 to 900 °C with a heating rate of 5 K/min under a normal atmosphere.

The geometric parameters and geometric porosity (P) of the light-frame honeycombs were determined with a caliper and on light microscopic images via ImageJ [37] on at least ten samples. The relative permittivity ε_r_ and the piezoelectric charge coefficient d_33_ were measured with the Berlincourt method (PM 300, Piezo Test, London, UK). Polarization was conducted with a corona poling set-up (1 kV/mm, 30 min) described in detail elsewhere [25]. The compressive strength was tested on at least ten samples per θ-angle along the Y-direction with a universal testing machine Instron 5565 (Instron Corp., High Wycombe, UK), a 500 N load cell, and 0.5 mm/min crosshead speed.

## 3. Results and Discussion

### 3.1. BCZT Sol–Gel Route

The sol–gel synthesis of the BCZT powder was based on a route reported by Castkova et al. whereas the more cost-efficient variant with carbonate precursors was chosen. However, isopropanol was replaced as a solvent with ethanol to reduce costs and process temperatures further. Instead of 100% isopropyl acetate with a boiling temperature of 89 °C, at least 50% ethyl acetate with a boiling temperature of 77 °C is formed during esterification, which additionally leads to a lower complexity of the ligands and thus to a lower decomposition temperature during calcination. A combined TGA DSC analysis, shown in Figure 3b, was performed on the dried xerogel for a more detailed investigation of the sequential reactions during the calcination process. Up to 300 °C (I), the mass decreases by 2.5%, due to the evaporation of the residual solvent, which is partly chemically bound. The further weight loss of about 5% up to 600 °C (II) is due to the decomposition of the acetate ligands to (Ba/Ca)_2_(Ti/Zr)_2_O_3_ CO_3_ [29,31,38]. An exothermic reaction occurs in the DSC curve at 620 °C (III). Here, the decomposition of the oxycarbonates starts under the release of CO_2_, with simultaneous crystallization of the BCZT phase [39]. This is accompanied by a further 15% loss of mass, which is completed at 760 °C (IV) and indicates complete crystallization at this temperature. Thus, by using ethanol instead of isopropanol, crystallization could be lowered almost by 200 °C, compared to Castkova et al. [29]. To verify phase purity, the XRD analyses, shown in Figure 3a, were conducted, and compared to the literature [29,40]. At a calcination temperature of 600 °C, a strong amorphous component is still evident, whereas from 650 °C phase-pure BCZT can be observed. This would confirm the assumption made in the DSC measurement that crystallization begins at 620 °C. However, the pure density measured by helium pycnometer at 650 °C revealed a density of only 4.48 g/cm^3^. With increasing calcination temperature up to 800 °C, the density increased to the expected density of 5.79 g/cm^3^ (theoretical density 5.82 g/cm^3^ [41]). This can be seen in the slight change in the XRD at the (100) peak where a reduction of amorphous oxycarbonate phase occurs. As can be seen from the TGA, decomposition and crystallization are therefore only completed above 760 °C. This highlights the urgency of analyzing sol–gel synthesized powders from several aspects to ensure phase purity.

The powder calcined at 800 °C was used for further tests and the manufacturing of light-frame honeycombs and replica foams. The powder has a specific surface area of 14.65 m^2^/g, and a mean particle size of 71 nm, calculated from the specific surface area. This is also consistent with the findings of Castkova et al. whose BCZT powders had a specific surface area of 11.25 m^2^/g and mean particle size of 98 nm at the same calcination temperature [29]. The change from isopropanol to ethanol shows therefore no significant effect on the resulting powder and is consequently a well-suited alternative.

### 3.2. Macro- and Microstructure of Light-Frame Ceramic Honeycombs

As shown in Figure 4a, it was possible to fabricate light-frame honeycombs from three different materials via the combination of stereolithography and the replica process. Important for the properties is the homogeneous strut diameter, which was well achieved except for the corner points and the open windows between the struts. Due to the surface tension of the slurry, rounding and material accumulation occurred at the internal angles, especially at negative angles (<0° auxetic), which could contribute to a reduction of stress peaks. This effect appeared independently of the material system. The residual strut pore shape after organic burn-out, which is shown in Figure 3b, also has a critical influence on the mechanical properties. Due to stress peaks at sharp corners in the pore, these are normally the most failure-prone areas, especially in replica foams from polyurethane templates. However, the triangular polymer strut shape used in this work transforms into a rounded triangle pore that leads to the reduction of stress peaks and thus exceeds the mechanical properties compared to standard replica foams as already shown in previous publications [25,42]. In addition, it has been shown that this rounded triangular shape is even superior to circular pores, as it favors a more homogeneous layer thickness [25].

The quantitative analysis of the structure is shown in Figure 5a based on angle and macro porosity. The lowest porosity was obtained at −33.8 ± 1.8° (−30°) with 81.3 ± 1.4 vol.% and rises than linearly to 84.9 ± 1.5 vol.% at 0.8 ± 0.1° (0°) and to the maximum at 29.3 ± 1.7° (30°) with 88.4 ± 1.3 vol.%. As expected, porosity increases with angle due to growing volume at the same weight. The angle could also be replicated precisely and shows an error of approx. 2° each. The only exception is the specimen at −30°, which deviates more clearly with 34°, caused by the material accumulation of the slurry in the sharp corners. Overall, the macro porosity can be adjusted very precisely via the geometry with a median standard deviation of 1.5% (corresponding to a weight deviation of ±0.02 g), although the infiltration process was handmade. The same values have already been determined in previous studies, which supports the reproducibility of the manufacturing accuracy [25,42]. In addition, the component geometry can also be printed directly by stereolithography, which means that no additional post-processing step is required after the replica has been molded.

### 3.3. Mechanical and Piezoelectric Properties of Light-Frame Ceramic Honeycombs

The mechanical properties of light-frame honeycombs were characterized by compression tests on Al_2_O_3_ samples. The maximum compressive strength is at −30° with 5.7 ± 2.1 MPa, halves to 2.6 ± 1.0 MPa at 0° and decreases further to 0.6 ± 0.4 MPa at 30°. The mean standard deviation of 47% is due to the variance of the porosity. The porosity has an exponential influence on the mechanical properties, which makes them sensitive to them. Nevertheless, standard deviations up to 50% are often seen in highly porous structures, especially at low strengths [7,25,42,44]. Therefore, despite high standard deviations, the strengths are representative for such porous cellular materials.

The result can be described using the model of Phani and Niyogi from Equation (1) [43], with 1 ≤ *a* ≤ 3.85 and 2.14 ≤ *n* ≤ 4.12. Parameter *a* is 1/P_critical_ where the strength becomes zero.
(1)$$\sigma_{c}=\sigma_{0}{\left(1-aP\right)}^{n}$$

The fitted model resulted in an *a* of 1.025 and *n* = 3.999 with an R^2^ of 0.98, which fulfills the parameter specifications and provides a very good estimation. The maximum strength of light-frame honeycombs was expected at an angle of 0°, as found in previous works on normal honeycombs [26,45]. However, they behave more like classical ceramic foams, whose mechanical properties are mainly influenced by porosity [6]. This could be attributed to the strut pores, which represent the largest defect in the component, thereby negating the positive influence of the structure. Nevertheless, the auxetic samples at −30° exceed the strengths of alumina replica foams, which are typically 0.5 to 3 MPa in the porosity range of 80 to 90 vol.% [44,46,47,48]. This shows that the combination of stereolithography and replica technology can improve the mechanical properties compared to the conventional process. Consequently, achieving the same strength at higher porosities is possible, which allows the light frame variant to save additional material and thus energy.

The piezoelectric properties were characterized in terms of d_33_ and relative permittivity ε_r_. However, due to the low strength of the BCZT light-frame honeycombs, they could not be measured because they already failed during the preload of the measurement. For reference, 30, 45, and 60 ppi replica foams were prepared from BCZT (sol–gel) in the same porosity range as the light-frame honeycombs, which could be measured. Since it was possible to produce and measure stable BT light-frame honeycombs and BCZT replica foams, the question arises why the BCZT light-frame honeycombs have insufficient mechanical strength. This effect can be attributed to the combination of the manufacturing process and the extremely small particle size of the BCZT powder. Since a template is coated several times in the replica process, the ceramic layers must dry without cracks. The drying process and the maximum crack-free layer thickness are strongly dependent on the particle size (h_max_ ≈ $R^{\frac{3}{2}}$) [49]. Due to the higher number of struts in the replica foam compared to the light-frame honeycomb, the layer thickness per infiltration cycle is significantly lower for the replica foam. While the particle size of 70 nm was suitable for producing replica foams without defects due to their thinner layer thickness, this was not the case for light-frame honeycombs. On the other hand, the use of deagglomerated BT powder containing particles of 2 µm enabled the production of mechanically robust light-frame honeycombs, corroborating the dependence of particle size. To produce BCZT light-frame honeycombs in the future, it would be necessary to increase the number of infiltration cycles. This would result in thinner intermediate layers for each infiltration step, thereby achieving the desired reduction in layer thickness and defect free drying.

Nevertheless, the piezoelectric charge coefficient d_33_ of BCZT sintered at similar conditions is 325 ± 26 pC/N [29], which is in the same range of the BT with 350 pC/N [50,51]. In addition, previous studies have shown that linear void structures of BT with a porosity of 59 vol.% achieved a d_33_ of 271 pC/N [25], while BCZT structures with 62 vol.% porosity attained a d_33_ of 296 pC/N [26]. Considering that BT and BCZT have comparable piezoelectric properties, it is possible to investigate the macrostructural influence in the following, independent of the material.

In Figure 6a, the piezoelectric coefficient d_33_ of the BT light-frame honeycombs and the BCZT replica foams is shown. The BT light-frame honeycombs achieved their maximum at −30° of 183.4 ± 29.7 pC/N with the lowest porosity of 64.4 ± 3.6 vol.%, and the minimum at 15° of 161.4 ± 23.5 pC/N with a porosity of 73.9 ± 1.9 vol.%. With increasing angle to 30° and porosity to 80.3 ± 1.8 vol.%, the d_33_ slightly increases to 176.6 ± 29.4 pC/N. However, due to the high standard deviations, no clear trend can be identified. Only a slight decrease in d_33_ with increasing porosity can be observed, as is also known from conventional piezoelectric foams [3,5,18,19].

The structural advantage of the light-frame honeycombs for the piezoelectric properties becomes clear in comparison with the BCZT replica foams, which are built up of random cells. The d_33_ for the 30 ppi foams with a porosity of 75.2 ± 2.7 vol.% is 62.1 ± 6.3 pC/N, 45 ppi (*p* = 69.9 ± 2.5 vol.%) is 58.0 ± 9.5 pC/N, and 60 ppi with a porosity of 59.2 ± 3.1 vol.% achieved 74.7 ± 11.6 pC/N. Thus, the BT light-frame honeycombs have a d_33_ that is higher by a factor of 2.5 despite their partially higher porosity, which reflects the enormous influence of the structure on the piezoelectric properties.

A value that highlights the structural difference is the specific surface area α_SS_, which can be calculated using Equation (2) with the surface area S and the total sample volume V. For a 45 ppi replica foam, it is α_SS_ = 12.0 × 10^3^ m^2^/m^3^ [52], for the BT light-frame honeycombs 3.2 × 10^3^ m^2^/m^3^, and for a normal BCZT honeycomb 1.7 × 10^3^ m^2^/m^3^ (calculated from data [26]). The specific surface area of the BT light frame honeycomb is increased by 1.88 times compared to the normal BCZT honeycomb, and that of the 45 ppi replica foam is increased by 3.75 times, relative to the light frame structure. If the d_33_ values of these three structures are included, an inverse relationship between d_33_ and specific surface area is obtained. Thus, the d_33_ of the normal honeycombs with the lowest specific surface areas is increased by 1.43 times compared to the light-frame honeycombs. The light-frame honeycombs, in turn, have a d_33_ 3.12 times higher than the 45 ppi foams, which have the highest specific surface areas. This inverse relationship between specific surface area and d_33_ can thus be expressed by Equation (3) and will be strengthened in the next section by the results of the permittivity and the determination of the depolarization coefficient N_i_. This can be determined from permittivity, described by Equation (4) with porosity P and the permittivity of the dense material ε_sl_ [53].
(2)$$\alpha_{SS}=\frac{S}{V}$$
(3)$$\frac{\alpha_{SS(1)}}{\alpha_{SS(2)}}\approx \frac{d_{33\left(2\right)}}{d_{33\left(1\right)}}\approx \frac{N_{i\left(1\right)}}{N_{i\left(2\right)}}$$
(4)$$\varepsilon_{r}=\frac{\left(1-P)(\varepsilon_{sl}-1\right)}{1+N_{i}*(\varepsilon_{sl}-1)}$$

Of the BCZT replica foams, those with 60 ppi show the highest relative permittivity at 165.6 ± 14.9. With increasing porosity and decreasing ppi number, the permittivity drops linearly to 107.5 ± 38.3 (45 ppi) and 94.3 ± 7.1 (30 ppi). The BT light-frame honeycombs achieved two times higher values with a maximum of 350.3 ± 63.6 at −30°. With simultaneously increasing angle and porosity the permittivity falls to 222.7 ± 43.3 at 30°. However, the lowest value occurs at the outlier 0° with 172.0 ± 64.2. Due to the random structure of the replica foams and the high specific surface, the electric field is strongly disturbed, which leads to a lower permittivity compared to the ordered BT light-frame honeycombs [25]. This in turn leads to an increase in the d_33_ values of the regular BT light-frame honeycombs, since the polarization is improved with increasing permittivity [54]. This is also reflected in the value of the depolarization coefficient N_i_, which is 0.00063 for the BT light-frame honeycombs and 0.00217 for the BCZT replica foams. The depolarization coefficient of the BCZT replica foams is thus higher by a factor of 3.4 compared to the BT light-frame honeycombs, which is approximately the inverse ratio of the d_33_ results and the same ratio of the specific surfaces. This is further confirmed by the normal (injection-molded) BCZT honeycombs, which have a N_i_ of 0.00032 [26], a depolarization factor 1.99 times smaller than that of the BT light-frame honeycombs. The relation between the specific surface area, piezoelectric charge coefficient and depolarizing factor can be found in Table 1. Consequently, through tailoring of the specific surface and therefore macro- and microstructure, the depolarization coefficient can be optimized to increase the piezoelectric charge coefficient. This also makes it possible to produce components with the same properties but higher porosity, which leads to savings in material and energy too.

## 4. Conclusions

In this work, novel light-frame honeycombs of BT, BCZT, and Al_2_O_3_ were successfully fabricated by combining stereolithography, replica and sol–gel processes. In addition, the BCZT powder was prepared via a sol–gel process and the effect of calcination temperature on powder properties was investigated. It was shown that there is a large energy saving potential for the synthesis of BCZT powder.

The influence of the macrostructure on the mechanical properties of the Al_2_O_3_ samples and the piezoelectric properties of the BCZT and BT structures were investigated. The Al_2_O_3_ light-frame honeycombs reached a maximum strength of about 6 MPa, which decreased with increasing porosity and structural angle. Phani’s model for estimating the strength based on porosity confirmed that the light-frame honeycombs were mechanically similar to replica foams. This was attributed to the hollow struts caused by the manufacturing process. To investigate the influence of the macrostructure on the piezoelectric properties, the piezoelectric charge coefficient d_33_ and the permittivity of BT light-frame honeycombs as well as BCZT replica foams were determined. Here, the light-frame honeycomb structures achieved a d_33_ above 180 pC/N, which is 2.5 times greater than the values for the replica foams. It has been shown that with reduced specific surface area, the d_33_ can be increased as a result of a smaller depolarization. This makes it possible to maintain consistent properties while reducing the amounts of raw materials, which in turn results in additional energy savings. It also allowed us to build a deeper understanding of how cellular structure and piezoelectric properties are related.

Combining the sol–gel process, replica technology, and stereolithography thus offers the possibility of producing highly complex geometries, which could lead to high mechanical and piezoelectric performance for highly sensitive sensors, hydrophones, or energy harvesting systems in combination with a soft polymer to form a flexible ceramic/polymer composite. Therefore, when designing components for piezoelectric applications, it is essential to consider both composition and microstructure in order to optimize the material and fabrication performance.

## Figures and Tables

**Figure 1 materials-16-04924-f001:**
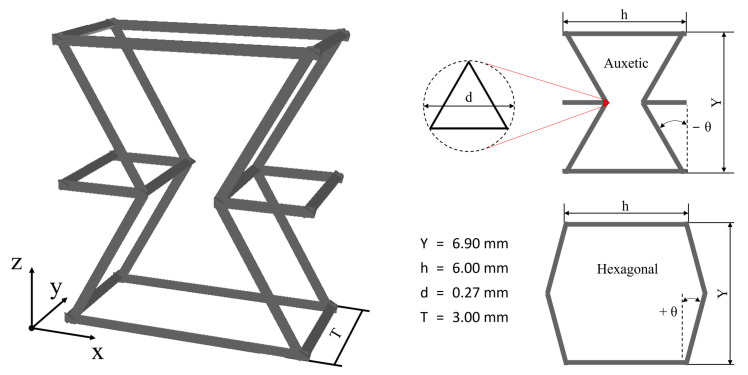
Light-frame auxetic (θ < 0°) and hexagonal (θ ≥ 0°) unit cell with structural parameters Y: height, h: width, d: strut diameter, T: thickness, θ: angle.

**Figure 2 materials-16-04924-f002:**
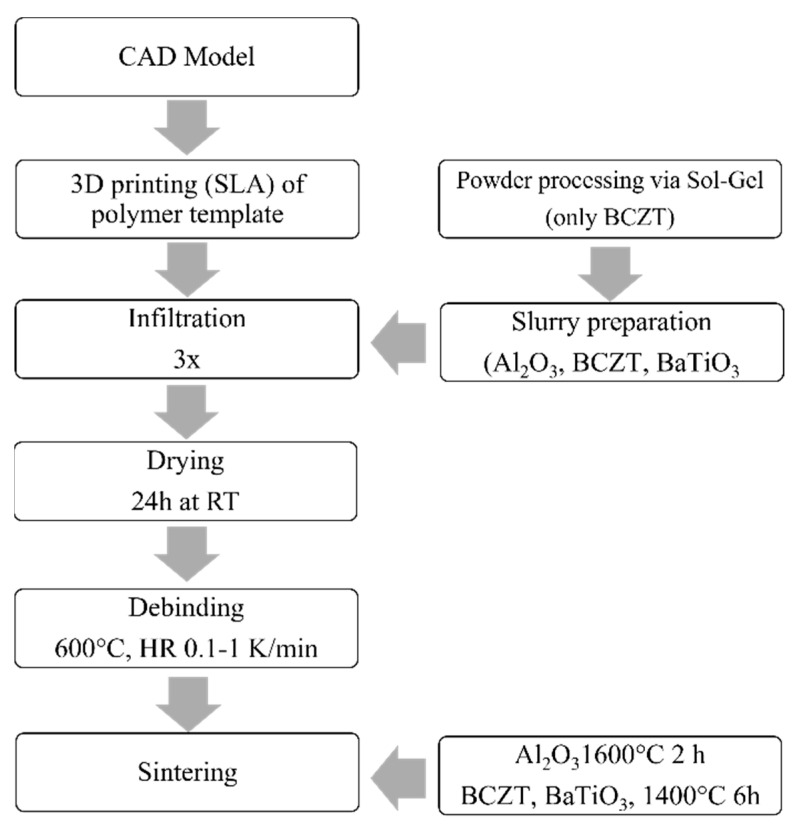
Process scheme of the manufacturing of light frame honeycomb structures, by combining 3D-printing, replica technique and sol–gel powder synthesis.

**Figure 3 materials-16-04924-f003:**
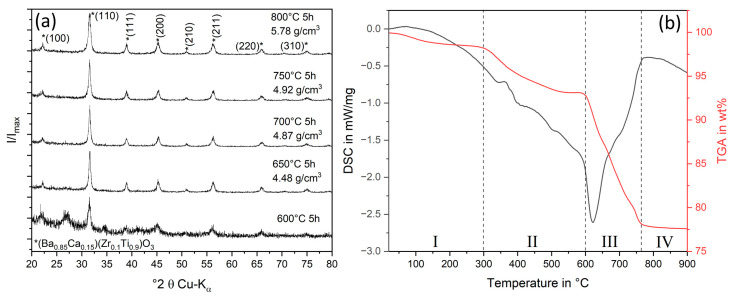
(**a**): XRD analysis of the BCZT *(Ba_0.85_Ca_0.15_)(Zr_0.1_Ti_0.9_)O_3_ powder calcined between 600 °C and 800 °C for 5 h and related density determined via helium pycnometer. (**b**): TGA/DSC measurement of the BCZT xerogel for calcination with the different temperature reaction zones I–IV.

**Figure 4 materials-16-04924-f004:**
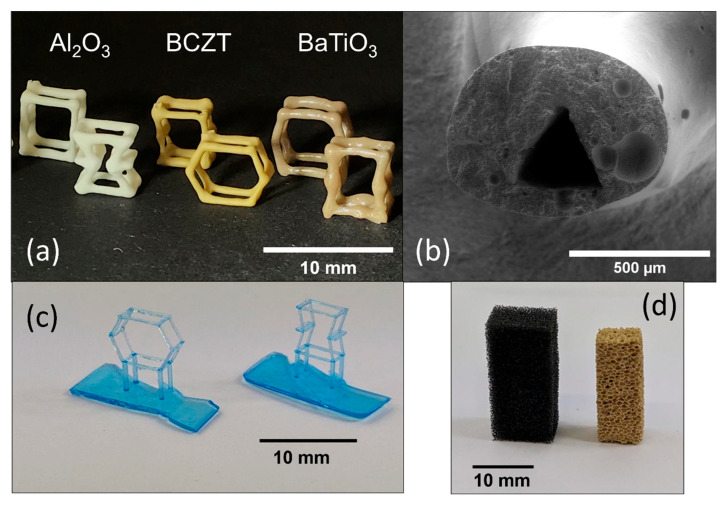
(**a**) Sintered light-frame ceramic honeycombs made of Al_2_O_3_, BCZT, and BaTiO_3_ with positive (hexagonal) and negative (auxetic) angles; (**b**) SEM image of a cross-sectional area of an Al_2_O_3_ strut with triangular pore; (**c**) 3D-printed polymer templates used in replication process (**a**); (**d**) Reference PU-Template and sintered BCZT replica foam from PU-Template.

**Figure 5 materials-16-04924-f005:**
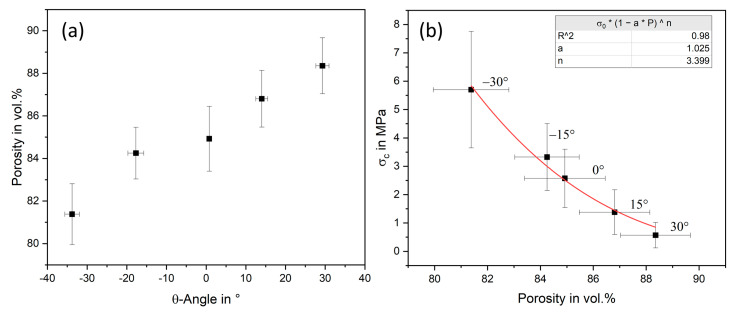
(**a**) Porosity in dependence of the structural angle θ of Al_2_O_3_ light-frame honeycombs; (**b**) compressive strength of Al_2_O_3_ light-frame honeycombs as a function of the porosity with a fitted model from Phani et al. [43].

**Figure 6 materials-16-04924-f006:**
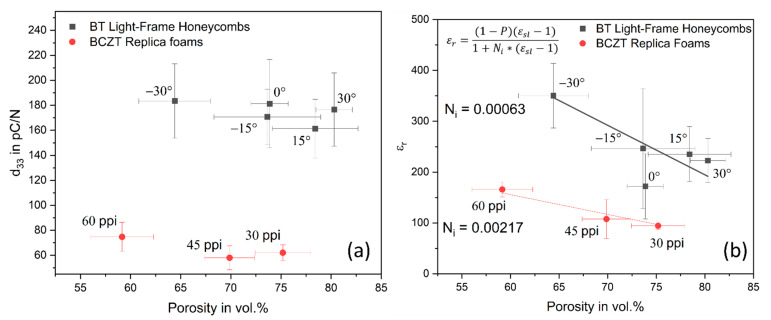
Piezoelectric properties of BT light-frame honeycombs and BCZT replica foams in dependence of the porosity; (**a**) piezoelectric charge coefficient d_33_; (**b**) relative permittivity with the fitted model from Okazaki.

**Table 1 materials-16-04924-t001:** Material parameters of 45 ppi replica foam and honeycombs (HC), specific surface (α_SS_), piezoelectric charge coefficient (d_33_), depolarizing coefficient (N_i_), and their ratios as marked by the indices (1) and (2) in color. Specific surface for the 45 ppi replica foam from [52], and properties for normal honeycombs from [26].

		α_SS_	d_33_	N_i_	α_SS(1)_/α_SS(2)_	d_33(2)_/d_33(1)_	N_i(1)_/N_i(2)_
		in m^2^/m^3^	in pC/N				
	45 ppi	12.0 × 10^3^	58.0	2.17 × 10^−3^	3.75	3.12	3.44
(1)	(BCZT)						
(2)	Light-frame HC	3.2 × 10^3^	181.3	0.63 × 10^−3^			
(1)	(BT, θ = 0°)				1.88	1.43	1.99
(2)	Normal HC	1.7 × 10^3^	259.0	0.31 × 10^−3^			
	(BCZT, θ = 0°)						

## Data Availability

The data presented in this study are available on request from the corresponding author. The data are not publicly available due to privacy reasons.

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
