# Peer review of "Energy-Reduced Fabrication of Light-Frame Ceramic Honeycombs by Replication of Additive Manufactured Templates"

_materials, 2023, doi:10.3390/ma16144924_

Round 1
Reviewer 1 Report
1. It would be better to show a full process concept diagram in CH 2.2.
2. 1) You should show a picture of the printed frame used in Figure 3a. You also need to indicate whether Figure 3a is before or after sintering, and show both the before and after images.
2) Also, it seems difficult to see that the auxetic and hexagonal designs presented in Figure 1 is applied to Figure 3a. This needs to be explained.
3) During the replication process, the precision of the structure and the uniformity of the struts appear to be very low due to material aggregation and flow. This should be accompanied by quantitative measurement and evaluation, and validation that the results are reproducible.
3. What is the viscosity of the ceramic sol-gel state used?
4. Is the porosity in Figure 4a a measured value for the entire structure? You need to explain how many repetitions this is and how you measured it.
5. Fig 5 is hard to understand. Both BT and BCZT appear to have been made with 3d printing-based replicas. By the way, why is BCZT written in PPI and not in angle?
- In the case of BCZT's light-frame, it seems that the experiment was not possible due to the low strength, and if so, the shape of the corresponding replica structure needs to be presented.
- You should also consider whether it's appropriate to call it replica foam.
Author Response
Reviewer 1
1) It would be better to show a full process concept diagram in CH 2.2.
Thank you for your comment. We added a figure (Fig. 2).
2) You should show a picture of the printed frame used in Figure 3a. You also need to indicate whether Figure 3a is before or after sintering, and show both the before and after images.
We modified Fig 4. and added necessary information to ensure a clear understanding for the reader. Since there is no visible change before and after sintering expect a slight change in color we refuse to show almost same picture twice.
3) Also, it seems difficult to see that the auxetic and hexagonal designs presented in Figure 1 is applied to Figure 3a. This needs to be explained.
We added more information, but due to the high amount of variations we decided to show only few not to confusing the reader.
4) During the replication process, the precision of the structure and the uniformity of the struts appear to be very low due to material aggregation and flow. This should be accompanied by quantitative measurement and evaluation, and validation that the results are reproducible.
We know this issue, which is described in 3.2 and this is reflected in the porosity distribution (4a) as well in the mechanical test. The porosity is mainly effected by this material aggregation. We described their influence on the mechanical properties in detail in the material properties section. We agree, that it might be interesting to know the amount of aggregation, but this will not gain more valuable information for the mechanical properties on our side.
5) What is the viscosity of the ceramic sol-gel state used?
Since we’re only interested in the powder derived from sol-gel, we didn’t measured the viscosity.
6) Is the porosity in Figure 4a a measured value for the entire structure? You need to explain how many repetitions this is and how you measured it.
We added some more information and explained it already in the methods section, which we pointed out more.
7) Fig 5 is hard to understand. Both BT and BCZT appear to have been made with 3d printing-based replicas. By the way, why is BCZT written in PPI and not in angle?
We clarified first the structures in Fig 4 by adding additional information in the figure description. The BT light frame structures are based on 3D printed replica templates and the BCZT is based on a replica foam. Therefore BCZT is in ppi.
In the case of BCZT's light-frame, it seems that the experiment was not possible due to the low strength, and if so, the shape of the corresponding replica structure needs to be presented.
That is right and mentioned in the text, the mechanical strength however was too low, so we changed to replica foam. We added the shape in figure 4.
You should also consider whether it's appropriate to call it replica foam.
We added a picture to clarify this. Thank you very much for your valuable comment.

Reviewer 2 Report
Review of materials-2462007, entitled ‘Energy-Reduced Fabrication of Light-Frame Ceramic Honey-2 combs by Replication of Additive Manufactured Templates’
This manuscript light-frame ceramic honeycombs synthesized with the combined sol-gel process, replica technology, and stereolithography techniques. The characterization of crystal structure, BET, microstructure, DSC-TGA, as well as mechanical and piezoelectric properties were systematically characterized. It is well organized. It is recommended to accept for publication after revising in the following aspects:
--In Figure 2, the increase in density was observed after calcination at 800oC while no phase change or DSC peak was traced. As well accepted, larger density is expected for amorphous material compared to the crystalized state.
--The author built a beautiful correlation between compressive strength and porosity, while the standard deviation of both strength and porosity is too large. Is the averaged value representative or not? More discussion is recommended.
Minor editing of English language is recommended.
Author Response
Reviewer 2
1) In Figure 2, the increase in density was observed after calcination at 800oC while no phase change or DSC peak was traced. As well accepted, larger density is expected for amorphous material compared to the crystalized state.
Thank you very much for this comment. Since the sintered density at 1500°C is in the same range, we carefully checked the XRD data again and found differences which explain the increase in density.
2) The author built a beautiful correlation between compressive strength and porosity, while the standard deviation of both strength and porosity is too large. Is the averaged value representative or not? More discussion is recommended.
Thank you very much for your valuable comment. We discussed it more and explained it.

Reviewer 3 Report
Generally, the work is good and the results are interesting. The research methods selected by the authors are adequate to the subject of the paper. The paper requires some improvements which are proposed below:
Please add reference numbers conforming JCPDS/ICDD files.
What was the reason for these investigations? There is no answer to this question.
Figures 4 and 5 show very large measurement errors, please explain.
What is the potential use proposed for such materials?
The topic of the paper is very interesting and worth publication.
I recommend the article for publication after a minor revision.
Author Response
Reviewer 3
1) Please add reference numbers conforming JCPDS/ICDD files.
There is no JCPDS/ICDD File. We added the reference to the paper, where the crystal structure is described.
2) What was the reason for these investigations? There is no answer to this question.
We added two sentence and pointed out the information to make it more clear.
3) Figures 4 and 5 show very large measurement errors, please explain.
We added the explanation in the text to make it more clear, that high standard deviations depend on the porosity.
4) What is the potential use proposed for such materials?
We added some information in the conclusion.
